# Optimal water use strategies for mitigating high urban temperatures

Bin Liu[1,2,3], Zhenghui Xie[1,3], Shuang Liu[4], Yujing Zeng[5], Ruichao Li[1,3], Longhuan Wang[1,3], Yan Wang[1,3], Binghao Jia[1], Peihua Qin[1], Si Chen[1,3], Jinbo Xie[1], ChunXiang Shi[6]

[1]State Key Laboratory of Numerical Modeling for Atmospheric Sciences and Geophysical Fluid Dynamics, Institute of Atmospheric Physics, Chinese Academy of Sciences, Beijing, China
[2]School of Software Engineering, Chengdu University of Information Technology, Chengdu, China
[3]University of Chinese Academy of Sciences, Beijing, China
[4]Key Laboratory of Mountain Hazards and Earth Surface Processes, Institute of Mountain Hazards and Environment, Chinese Academy of Sciences, Chengdu, China
[5]Program in Atmospheric and Oceanic Sciences, Princeton University, Princeton, New Jersey, USA
[6]National Meteorological Information Center, China Meteorological Administration, Beijing, China

*Correspondence to*: Zhenghui Xie (zxie@lasg.iap.ac.cn)

**Abstract.** Urban irrigation and road sprinkling are methods for mitigating high urban temperatures, which are expected to enhance evapotranspiration and affect the urban weather, climate, and environment. Optimizing limited water supplies is necessary in regions with water shortages. In this study, we implemented urban water usage schemes including urban irrigation and road sprinkling in the Weather Research and Forecasting (WRF) model, and assessed their effects with different amounts of water in city centers, suburbs, and rural areas by using the WRF model at a resolution of 1 km in Beijing, China. In addition, we developed an optimization scheme with a cooling effect as the optimal objective and the total water supply as the constraint condition. Nonlinear relationships were identified between the cooling effect and water consumption for both road sprinkling and urban irrigation, and the cooling effect due to urban irrigation was more effective than that attributed to road sprinkling. Based on the optimal water management scheme and according to Beijing's 13th Five Year Plan, about 90% of the total water supply should be used for urban irrigation and 10% for road sprinkling as the most effective approach for decreasing urban temperatures by about 1.9°C.

## 1. Introduction

Throughout the world, the level of urbanization increased from 39% to 55% in the last four decades (Chen et al., 2014). Vast numbers of people have moved from rural to urban areas across the world,

thereby increasing greenhouse gas emissions, anthropogenic heat flux release, and energy consumption in urban areas, as well as causing land use and land cover changes that increase the likelihood of urban high temperature events (McCarthy et al., 2010). The frequency of high temperature events in the first decade of the 21st Century was much higher compared with that in the last 10 years of the 20th Century (WMO, 2013). According to a previous report, the temperature reached a maximum of 42.1°C in Beijing, China on August 3, 2018. In addition, increased temperatures can substantially increase the rate of temperature-related illnesses (Zhang et al., 2014). Applying water can cool urban areas directly by increasing transpiration from vegetation and evaporation from the soil (Coutts et al., 2013). Beijing is a region that lacks adequate water resources and optimal water use strategies can help to improve the water cooling efficiency. Thus, understanding and quantifying the relationships between the amounts of water applied and the cooling effect are critical for designing and planning better cities.

Urban irrigation includes ecological irrigation in city centers and farmland irrigation in suburban and rural areas. Many previous studies focused on the impacts of urban irrigation on hydro-meteorological variables at different scales (Kueppers et al., 2007; Vahmani and Hogue, 2014). Clearly, irrigation is a critical component of the regional water cycle because it enhances evapotranspiration due to the increased soil moisture contents and it contributes substantially to the latent heat flux in land–atmosphere interactions (Coutts et al., 2013; Pei et al., 2016). This so-called "oasis effect" is common in arid and semiarid cities. The impacts of irrigation on precipitation depend on the atmospheric circulation, where increasing the soil moisture can increase rainfall ( Moore and Rojstaczer, 2001; DeAngelis et al., 2010; Alter et al., 2015; Pei et al., 2016 ; Yang et al., 2017), whereas it may inhibit rainfall in other cases due to the evaporative cooling effect strengthening the atmospheric stability and weakening deep convection (Ek and Holtslag, 2004; Zeng et al., 2017). In addition, outdoor water usage changes the partitioning of the available energy between the sensible and latent heat fluxes. A decrease in the sensible heat flux can reduce the urban air temperature by more than 3 ℃, which helps to reduce thermal stress in cities during the summer (Kueppers et al., 2007; Lobell et al., 2008; Puma and Cook, 2010; Mueller et al., 2016). However, the cooling effects of different irrigation distributions differ slightly. The reductions in the daily maximum air temperature due to irrigation are evident in all urban land use types, but well vegetated and low intensity residential areas such as suburbs and rural areas exhibit the largest effects (Gao and

Santamouris, 2019). As for urban, the optimal distribution of water supplies to mitigate urban high temperatures in the summer is a problem. Indeed, optimizing water usage is limited by the agricultural water demand, crop production, and water transactions ( Feinerman et al., 1985; Amir and Fisher, 1999; Kuschel-Otarola et al., 2018). However, an effective method is not available for determining the distribution of the water supply to achieve the optimal cooling effect while also meeting the minimal requirements for plants.

Sprinkling water on the road can keep roads clean and control air pollution, and it is also an effective method for mitigating urban high temperatures and the urban heat island effect (Yamagata et al., 2008; Hendel and Royon, 2015; Hendel et al., 2016). However, the relationship between the amount of water applied by road sprinkling and the cooling effect in different urban areas has not been investigated. Moreover, urban irrigation and road sprinkling have different roles in the water cycle process, where urban irrigation is related to plant and soil processes, whereas road sprinkling responds directly to the atmosphere. Thus, determining the different cooling effects of these two water usage approaches is essential for developing water management strategies to mitigate urban high temperatures.

In this study, we determined the optimal method for distributing water by urban irrigation and road sprinkling in different parts of a city in order to mitigate urban high temperatures. We elucidated the relationship between the amount of water applied and the cooling effects of urban irrigation and road sprinkling based on simulations with the Weather Research and Forecasting (WRF) model. We then investigated whether the proposed method can be applied to other cities. We also collected water usage data for Beijing based on the water deficit coefficient, water supply, and land use cover in our case study, and modified the urban irrigation and road sprinkling schemes in the urban canopy and hydrology modules of the WRF model before conducting simulations.

The remainder of this paper is organized as follows. In Section 2, we describe the materials and methods employed, including the model development, data, and experiments conducted. In Section 3, we present our results and discussion, including the model validation process, relationships between the amount of water applied and the cooling effect, and the optimal water use strategies. We give our concluding remarks in Section 4.

## 2. Materials and Methods

### 2.1 Model development

The WRF model is a limited area, nonhydrostatic, mesoscale modeling system with a terrain-following eta coordinate, which is coupled with land surface models and the Urban Canopy Model (UCM) to provide a better representation of the physical processes related to the exchange of heat, momentum, and water vapor in an urban environment. The land surface model describes the physical soil hydrological processes explicitly, including infiltration, storage, redistribution, drainage, and evaporation. The UCM is a single layer model with a simplified urban geometry, where its features include shadowing from buildings, reflection of short and long-wave radiation, the wind profile in the canopy layer, and multi-layer heat transfer equations for roof, wall, and road surfaces (Tewari et al., 2007). The impervious roads lack soil hydrological processes but the evaporation of liquid water still occurs above the road, which can change the urban weather, climate, and environment.

A simple urban water usage scheme including urban irrigation and road sprinkling was incorporated into the WRF model based on the scheme proposed by Zeng et al. (2017). Ecological and farmland irrigation were both treated as urban irrigation and implemented in the same manner in the model. The soil hydrological processes were changed and the water balance between the land surface and atmosphere was disturbed provided that the irrigation water was added to the first layer of the soil and it was regarded as the available liquid water in the model. This process was conducted for farmland and urban land use types, and the water added from the surface soil was removed from the ground water table to maintain the water balance. According to the Requirements for Quality and Operation of City Road Sweeping and Cleaning (NO: DB11/T 353-2014) (Beijing M. 2015) the road sprinkling scheme was activated in the night when water was applied to the impervious road layer to accelerate evaporation. A flowchart illustrating the urban water usage scheme, including irrigation and road sprinkling in the model, is shown in Figure 1, and the specific scheme is represented by Eqs (1)–(4). The advanced water usage scheme mentioned above was coupled into the WRF model. Water from urban irrigation with a specific spatial distribution was entered as an input for the model via a data interface with the WRF model. The program was initialized for the real-time case study and the amount of water applied for road sprinkling was no more than the maximum water-holding capacity of the urban impervious layer:

$$\mathrm{w}_{i,j,t} = \begin{cases} wi_{i,j,t} & ,pervious\ layer \\ a \times pondmax_{urban} & ,impervious\ layer \\ 0 & ,otherwise \end{cases} \tag{1}$$

$$\mathrm{W}_{i,j,t} = \mathrm{W}_{i,j,t-1} + \mathrm{w}_{i,j,t} \tag{2}$$

$$\mathrm{wo}_{i,j,t} = \begin{cases} wi_{i,j,t} & ,pervious\ layer \\ 0 & ,otherwise \end{cases} \tag{3}$$

$$\mathrm{zwt}_{i,j,t} = \mathrm{zwt}_{i,j,t-1} - \mathrm{wo}_{i,j,t}, \tag{4}$$

where the subscripts $i$, $j$, and $t$ represent the latitude, longitude, and time, respectively, $wi_{i,j,t}$ represents the amount of water from urban irrigation, $a$ is a coefficient from 0 to 1, $pondmax_{urban}$ is the maximum water-holding capacity of the urban impervious layer, which mainly refers to the impervious road in the present study, $\mathrm{W}_{i,j,t}$ is the surface liquid water entering the first soil layer or impervious layer, $\mathrm{W}_{i,j,t-1}$ is the previous time step before $\mathrm{W}_{i,j,t}$, $\mathrm{wo}_{i,j,t}$ is the amount of water that needs to be removed from the ground water table, $\mathrm{zwt}_{i,j,t}$ represents the water table at time $t$, and $\mathrm{zwt}_{i,j,t-1}$ represents the previous time step at $\mathrm{zwt}_{i,j,t}$.

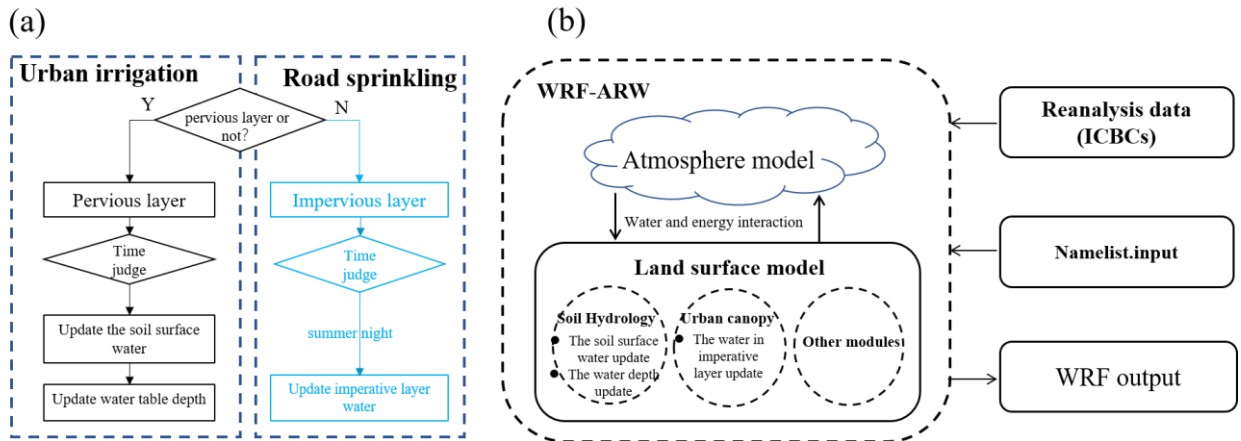

**Figure 1. Water usage scheme and its coupling with the Weather Research and Forecasting (WRF) model. (a) Flowchart illustrating the water usage scheme, including urban irrigation and road sprinkling. (b) Schematic showing the WRF model coupled with the water usage scheme.**

### 2.2 Data and experiments

Air temperatures obtained from reanalysis data and in-situ data were used to validate the WRF model output. In-situ data were obtained from 20 national meteorological stations in Beijing, and the regional reanalysis data came from the China Meteorological Administration Land Data Assimilation System (CLDAS) with hourly outputs with a resolution of 0.0625 ° × 0.0625 °. More detailed descriptions of the

data are given in Table 1. In addition, these data were also collected to verify the effectiveness of the WRF physical schemes.

**Table 1. Data set descriptions**

| Scale | Resource | Other details |
|-------|----------|---------------|
| Site | National meteorological stations | 20 observation sites; hourly; 2001–2017 |
| | Flux station | 140 m high at 39.9°N, 116.38°E, July to August in 2012 |
| Regional | CLDAS | $0.0625° \times 0.0625°$, hourly; 2008–2014 |

CLDAS: China Meteorological Administration Land Data Assimilation System (Shi et al., 2011)

The grid water usage data were derived from the total water consumption in Beijing and distributed according to the grid population density, GDP, and water deficit efficiency. The downscaling method employed was reported in a previous study (Zeng et al., 2017). The spatial distributions of the water usage amounts for the summed irrigation and ecological water are shown in Figure 2(a), and the spatial distributions of the road area proportions related to road sprinkling are shown in Figure 2(b). Farmland irrigation was mainly located in the south of Beijing and ecological water consumption occurred mainly in the center of the city. A primeval forest with little human influence is located to the north of Beijing and the water consumption was low in this area. The urban planning for Beijing can be separated into urban, suburban, and rural areas divided by the fifth and sixth ring roads. The area within the fifth ring road was treated as the urban area inhabited by the majority of the population. The population declined from the fifth to the sixth ring road in the so-called suburban transition area. The rural areas were located outside the sixth ring road, where they mainly comprised farmland with few building and factories.

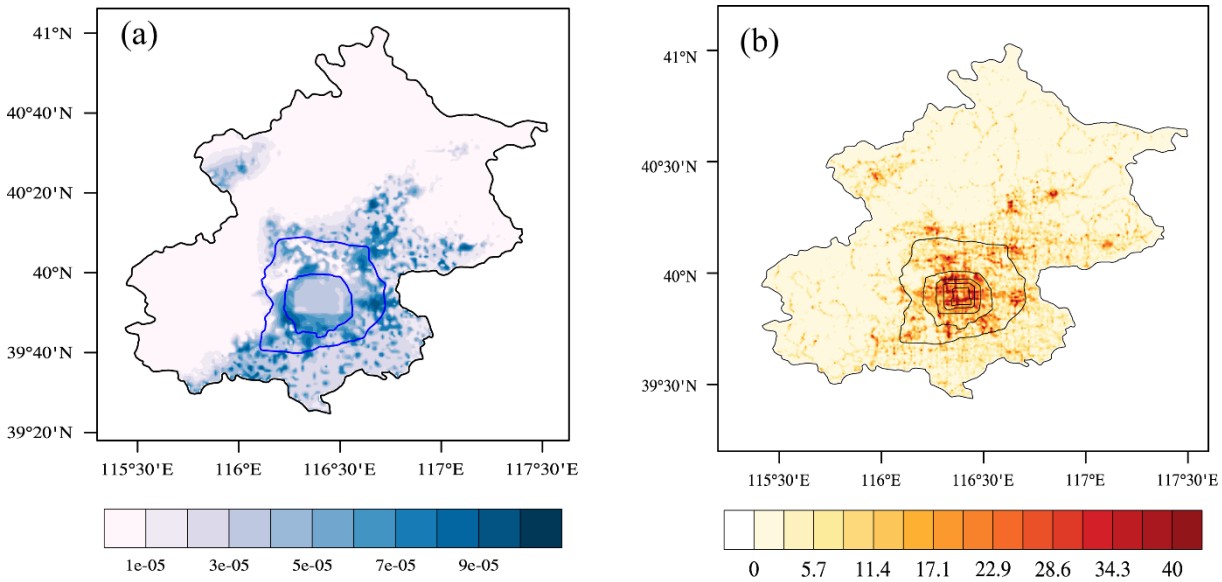

**Figure 2. (a) Estimated urban irrigation water use in Beijing (mm/s). (b) Spatial distribution of road area proportions (%).**

We used the WRF model (Skamarock, 2008) with the Advanced Research WRF Dynamic Core version 3.9.1 coupled with a single-layer UCM in the experiments. Three types of experiments were conducted to consider no water usage, urban irrigation, and road sprinkling in the city center, suburbs, and rural areas. The urban irrigation experiments comprised 21 individual model simulations, where the grid water usage data ranged from 0.1 to 1.9 times the estimated urban irrigation in the city center, suburbs, and rural areas. In each case, the data were regarded as new input variables and added to the initial model input files with the same spatial resolution as the model when the WRF model was running. The road sprinkling experiments comprised 27 individual model simulations, where the amount of water sprinkled on roads ranged from 0.2 to 1 times the maximum water-holding capacity of the impervious layer in three parts of the city with different urban sprinkling frequencies and strengths. Detail descriptions of the experimental designs are shown in Table 2.

**Table 2. Descriptions of experimental designs**

| Experiment | Area | Amount of water | Description |
|---|---|---|---|
| Raw experiment | / | / | No urban irrigation and no road sprinkling |
| Urban irrigation experiment | City center | 0.1, 0.4, 0.7, 1, 1.3, 1.6, and 1.9 times the estimated urban irrigation in each part of the city | Urban irrigation in city center with different amounts of water |
| | Suburban areas | | Urban irrigation in suburban areas with different amounts of water |
| | Rural areas | | Urban irrigation in rural areas with different amounts of water |
| Road sprinkling experiment | City center | 0.2, 0.3, 0.4, 0.5, 0.6, 0.7, 0.8, 0.9, and 1 times the maximum water-holding capacity of the impervious layer | Road sprinkling in city center with different amounts of water |
| | Suburban areas | | Road sprinkling in suburban areas with different amounts of water |
| | Rural areas | | Road sprinkling in rural areas with different amounts of water |

Three-layer nested domains with horizontal resolutions of 15 km (d01; mesh size 95 $\times$ 121, most of northern China), 5 km (d02; mesh size 135 $\times$ 185; almost all of the Jing-Jin-Ji metropolitan area), and 1 km (d03; mesh size 205 $\times$ 270; Beijing as the area of interest) were designed for the experiments (Figure 3). The National Centers for Environmental Prediction Global Final Analysis 6-h data (soil water, moisture, and temperature) were used for the first-guess initial field and lateral boundary conditions. The MODIS-based land use classifications data provided in the WRF model described the real terrestrial and land-cover characteristics of the regions of interest, and the default static data were used in the experiments. Climate summer time periods from 2000 to 2017 were averaged to 4 days to represent climatic May, June, July, and August. First, we obtained all of the data for May from 2000 to 2017, before averaging all these data to one day to represent climatic May. Finally, climatic June, July, and August were obtained by repeating these two steps. The first day (climatic May) was considered as the spin-up period. The schemes are shown in terms of the physical options in Table 3.

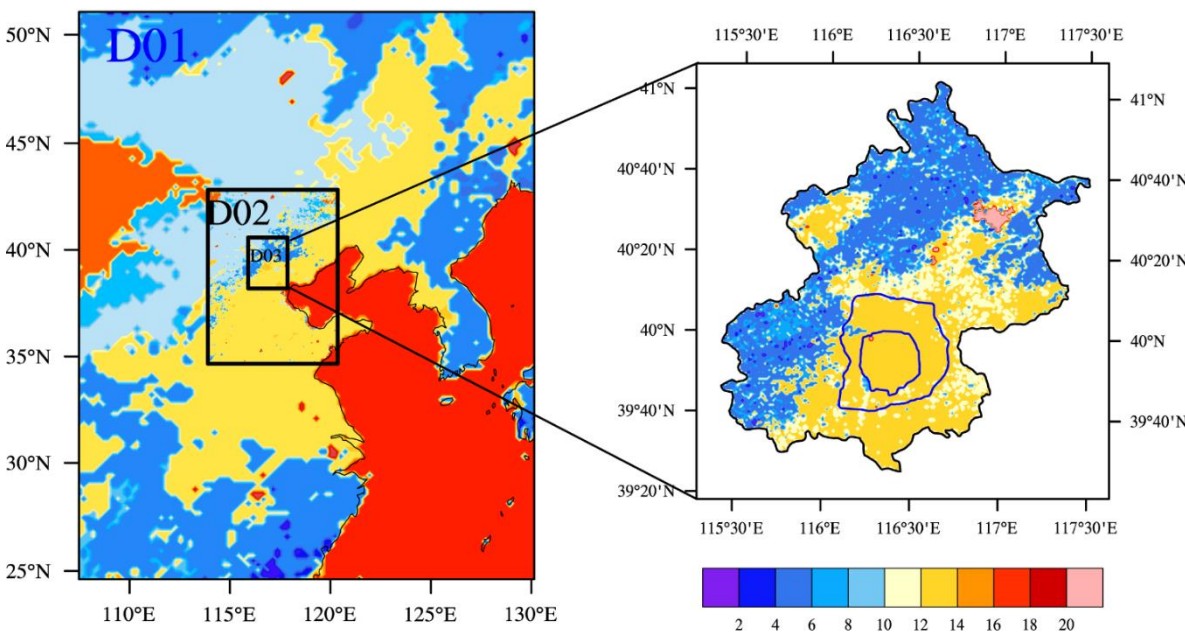

**Figure 3. Simulated area, land use, and land cover in Beijing.**

**Table 3. Physical parameterization schemes**

| Physical scheme | Selected scheme option |
| --- | --- |
| Microphysics | Kessler scheme |
| Longwave | RRTM scheme |
| Shortwave | MM5 shortwave scheme |
| Cumulus | Grell-Devenyi |
| Planetary boundary physics | ACM2 PBL scheme |
| Land surface model | NOAH-MP |
| Urban model | SLUCM |

## 3. Results and Discussion

### 3.1 Model Validation

Considering that random processes in the atmosphere may lead to uncertainty regarding the cooling effect, offline comparative experiments were conducted to understand the cooling effects of urban irrigation and road sprinkling. These experiments comprised raw simulations without urban water usage,

and with urban irrigation and road sprinkling using the community land model (CLM 4.5). The simulations were driven by ITP atmosphere forcing data (June to August in the year 2012) obtained from the Data Assimilation and Modeling Center for Tibetan Multi-spheres, Institute of Tibetan Plateau Research, Chinese Academy of Science (Yang et al., 2010a). The simulation results showed that urban irrigation decreased the groundwater table due to groundwater extraction, as also shown by Yang et al. (2010b), and the surface soil moisture increased (Figure 4Figure 4). There were no changes in the water table and surface soil moisture under road sprinkling due to the lack of underground water processes below the impervious layer. Evapotranspiration was strengthened when more water was applied to irrigate soil or farmland, and heat from the ground and air was taken away. Similar results were obtained under road sprinkling but the physical process was slightly different. As a result, both schemes decreased the sensible heat flux, ground temperature, air temperature, and wind speed, and increased the latent heat flux (Figure 5Figure 5). In addition, the impacts of land surface variables were limited to the areas where water was applied because the offline simulations did not consider climate effects between the atmosphere and land surface. Thus, the cooling effects were fairly obvious under urban irrigation and road sprinkling, as also shown in previous studies (Wang et al., 2019; Hendel and Royon, 2015). Moreover, road sprinkling has been conducted in Beijing and Tokyo. In addition, the sensible heat flux and latent heat flux results obtained from the urban water usage simulations (including both urban irrigation and road sprinkling) were better than the raw model results (Figure 6Figure 6). Flux station observations from July to August in year 2012 and the station simulation results were interpolated from the regional simulation results in 2012. Comparisons of these data showed that the correlation coefficient increased slightly and the root mean squared errors for sensible heat flux and latent heat flux decreased by 4.69 and 6.94 $W/m^2$, respectively, while the absolute errors for sensible heat flux and latent heat flux decreased by 7.3 and 9.62 $W/m^2$ . Therefore, urban water usage, including urban irrigation and road sprinkling, should be considered when conducting weather and climate simulations.

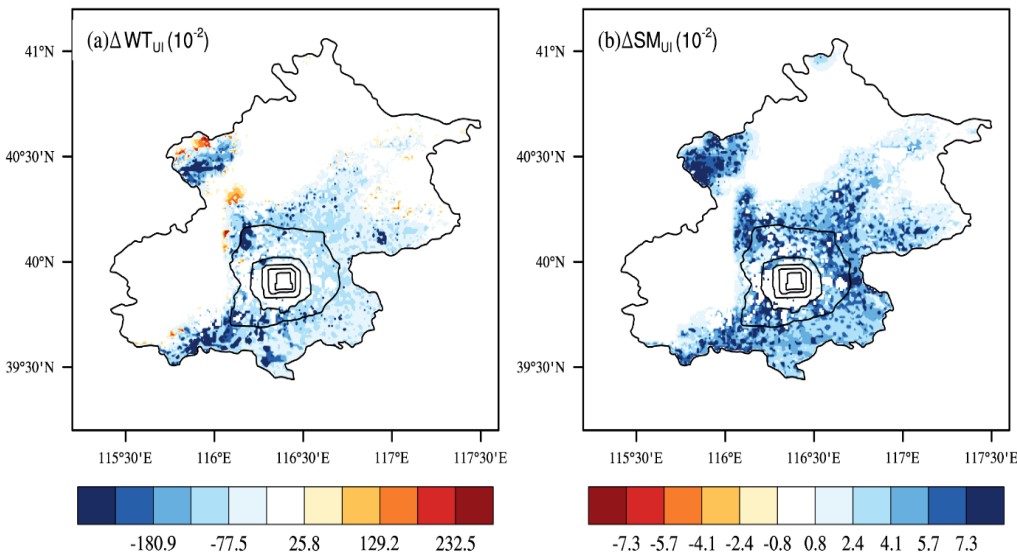

**Figure 4. Changes in (a) ground water table and (b) surface soil moisture due to urban irrigation.**

215

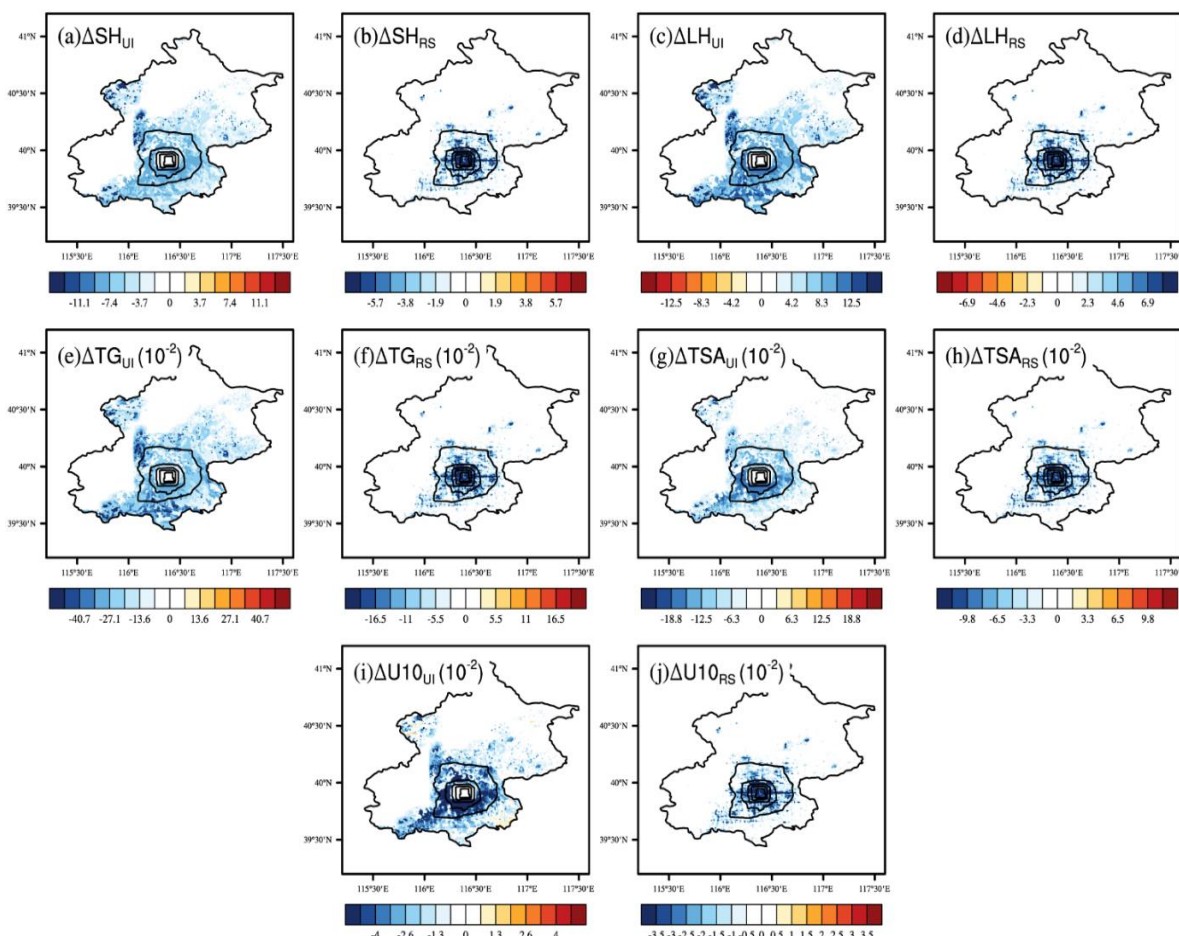

**Figure 5. Changes in land surface variables due to urban irrigation (UI) and road sprinkling (RS).** (a) Sensible heat flux under UI, (b) sensible heat flux under RS, (c) latent heat flux under UI, (d) latent heat flux under RS, (e) ground temperature under UI, (f) ground temperature under RS, (g) air temperature under UI, (h) air temperature under RS, (h) 10-m wind speed under UI, and (j) 10-m wind speed under RS.

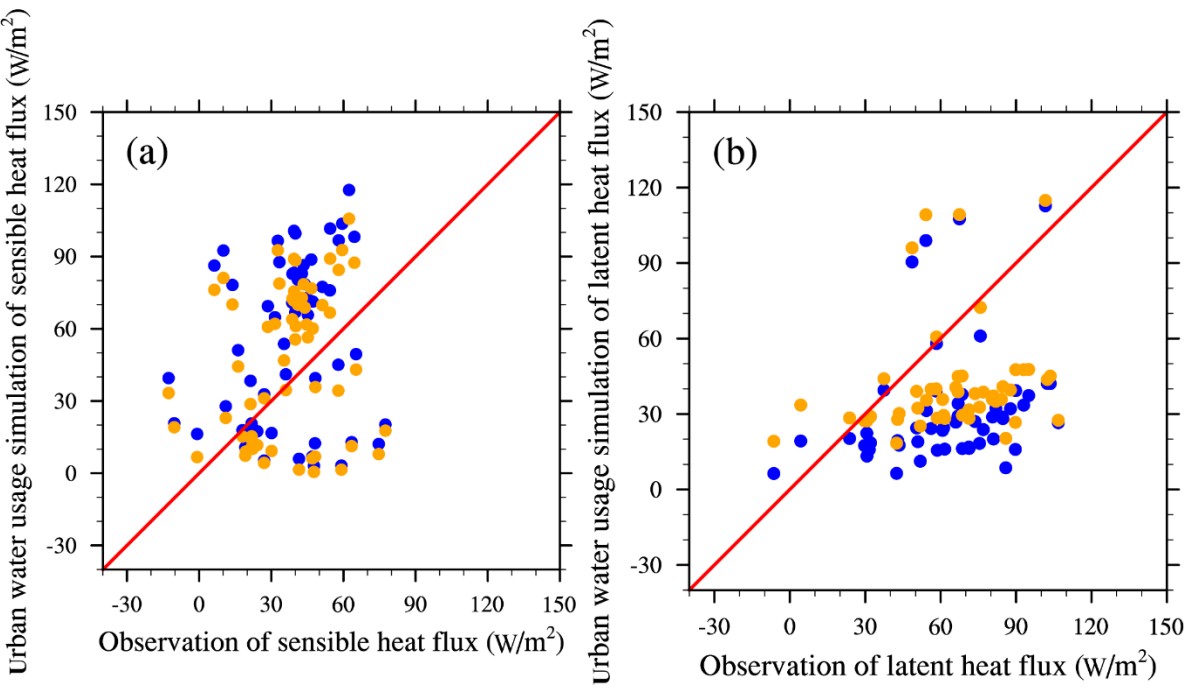

**Figure 6. (a) Comparisons of sensible heat flux and (b) latent heat flux according to station observations and CLM simulations. Blue dots are the raw CLM simulation results and orange dots are the CLM simulation results with urban water usage scheme.**

In addition to the comparisons of station observations and offline model simulations, we conducted comparisons of the 7-year average summer temperatures in the CLDAS and WRF simulations to evaluate the simulation capacity of the raw WRF model, where the temperatures were higher in the city center than the suburbs. The similar spatial distributions showed that the WRF physical scheme was reasonable. The correlation coefficients between the CLDAS temperatures and WRF simulation results were generally close to one, and the average root mean square error (RMSE) for the two data sets was 0.8°C. The grid model results were interpolated to the sites according to the coordinates of the stations in urban and suburban areas, and comparisons of the site observations and WRF simulation results also showed that the temperature simulation results were in good agreement with the observations (see Figure 7). The WRF model simulation results were reasonable, where the simulated temperatures were slightly higher than the observations and RMSE was mostly less than 2.5°C. Thus, the WRF model may be a suitable tool for this type of research.

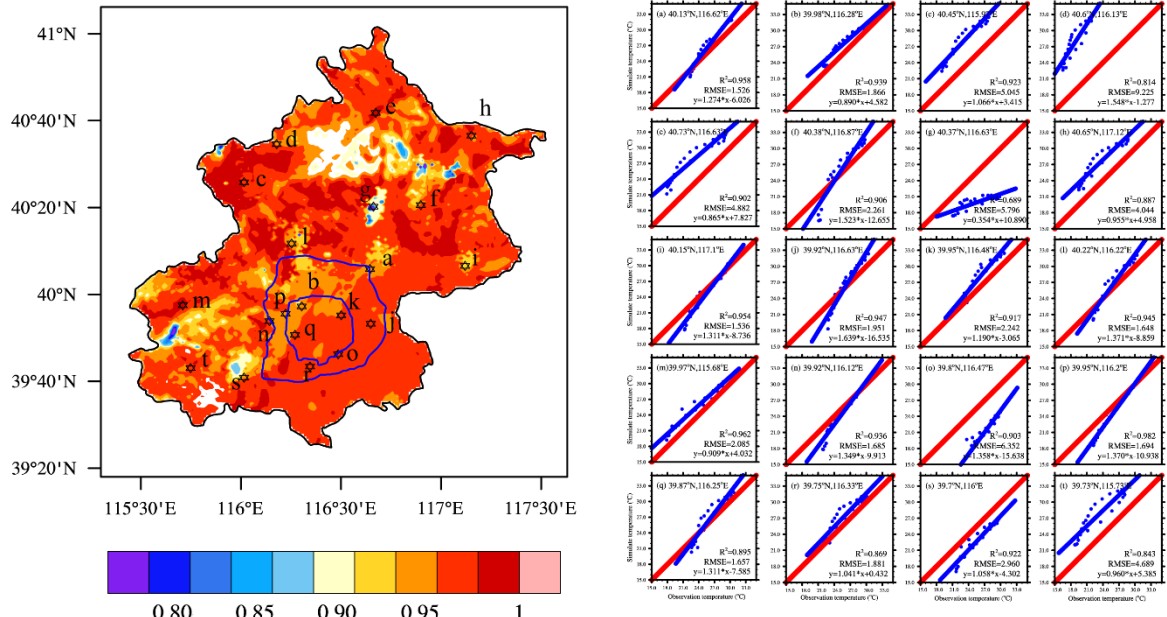

**Figure 7. Temperature validation based on comparisons of simulations and reanalysis or in-situ data. Left: spatial distribution of correlation coefficients between simulations and CLDAS reanalysis data. Right: (a)–(t) regression lines, RMSE values, and coefficient coefficients between simulations and in situ data.**

Also, WRF simulations with and without urban water usage schemes were conducted from July to August, 2012. The comparisons of the reanalysis data (CLDAS) and station observations showed that the simulations with urban water usage were better than the raw WRF simulations in terms of the heat flux, and temperature (Figure 8). The mean absolute error of the air temperature decreased from 2.9 to 1.7°C. compared with the CLDAS reanalysis data. Compared with the station observations, the correlation coefficients for sensible heat flux from simulations with and without water usage schemes changed little, and correlation coefficients for latent heat flux increased by 0.07, the root mean square errors decreased from 57.6 to 52.6 $W/m^2$ after applying urban water usage schemes.

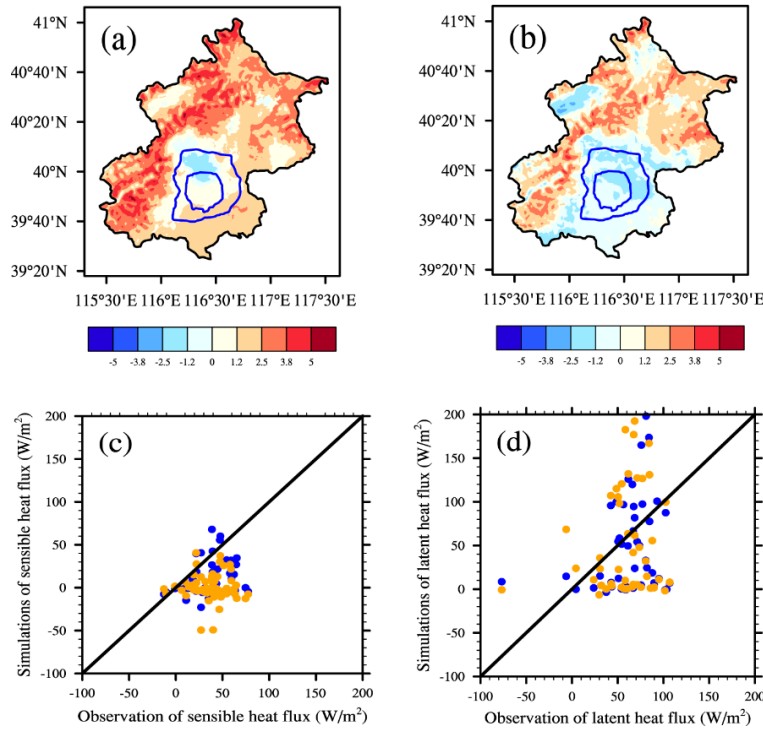

**Figure 8. Comparisons between WRF simulations with and without water usage schemes. (a) temperature comparisons between CLDAS and WRF model simulations without urban water usage scheme, (b) temperature comparisons between CLDAS and WRF model simulations with urban water usage scheme, (c) sensible heat flux comparisons between station observation and WRF simulation, (d) latent heat flux comparisons between station observation and WRF simulation. Blue dots are the raw simulation results and orange dots are the simulation results with urban water usage schemes.**

## 3.2 Relationships between the amounts of water applied and the cooling effect

Figure 9 shows that road sprinkling and urban irrigation both decreased the air temperature. Road sprinkling was mainly conducted at night to avoid disturbing traffic. The simulation results in Figure 9 show that the temperature decreased by a maximum of around 0.5–1°C in the city center where most roads were found, whereas there were no significant decreases in the temperature in rural areas where the lowest amount of road sprinkling occurred due to the low quantity of road surfaces in these areas. However, urban irrigation during the daytime decreased the temperatures in the day and night. Figure 10 shows that urban irrigation in the city center decreased the temperature by more than 1°C when large

volumes of water were applied. In the rural areas, the water applied for farmland irrigation had a reasonable cooling effect in the daytime, and the cooling effect continued but it was smaller in the nighttime due to the evaporation from farmland crops and urban plants after irrigating in the daytime. This effect was much more significant in the rural areas than the city center and suburbs. In addition, localized water usage could influence all of the areas, where road sprinkling or urban irrigation in the city center could decrease the temperatures in the suburbs or rural areas, and this effect may also occur in other locations. In general, the cooling effect of urban irrigation was stronger than that of road sprinkling because the amount of water applied for urban irrigation was greater than that for road sprinkling.

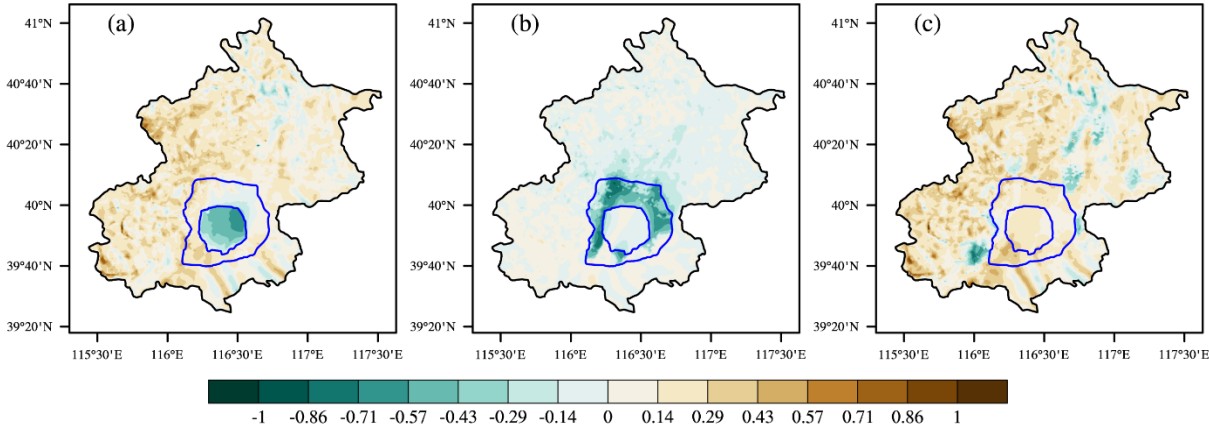

**Figure 9. Cooling effects in the city center (a), suburbs (b), and rural areas (c) due to road sprinkling in the night. The amount of water was half the maximum water-holding capacity of the urban impervious layer.**

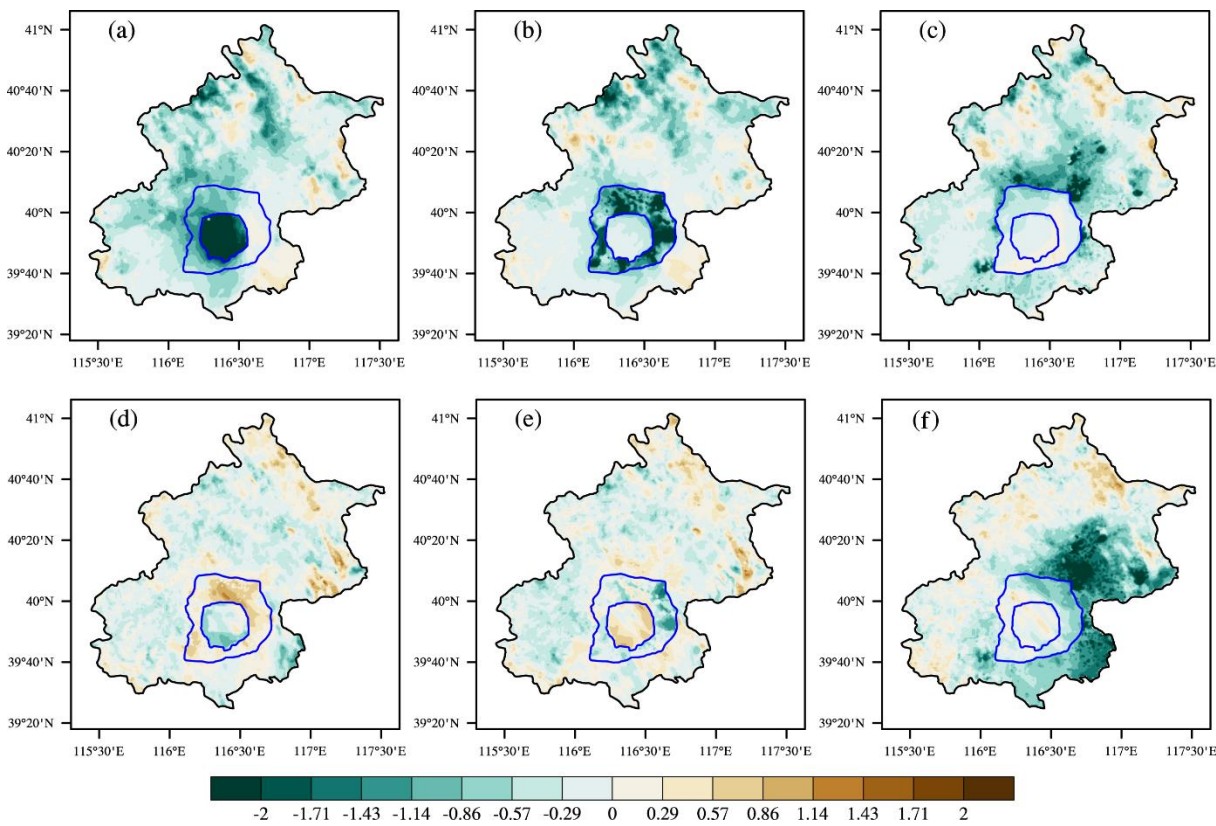

**Figure 10. Cooling effects in the city center, suburbs, and rural areas due to urban irrigation during the day and night. (a) Cooling effect of urban irrigation in city center during the day. (b) Cooling effect of urban irrigation in suburbs during the day. (c) Cooling effect of urban irrigation in rural areas during the day. (d) Cooling effect of urban irrigation in city center during the night. (e)**
**Cooling effect of urban irrigation in suburbs during the night. (f) Cooling effect of urban irrigation in rural areas during the night. The amount of water was the estimated urban irrigation.**

The application of water in urban areas could change the energy cycles and dynamic processes. Figure 11 shows that the changes in these variables were most significant in the areas where road sprinkling or urban irrigation were conducted and they could weaken the dynamic atmospheric processes.
The application of water by road sprinkling or urban irrigation increased the latent heat flux, decreased the sensible heat flux, and lowered the boundary layer heights locally. However, changes could be more general throughout the whole region. For example, urban irrigation in the city center lowered the boundary layer height in the city center, but also affected the suburbs and rural areas. Similar results were found for the latent heat flux and sensible heat flux, but they were not as significant.

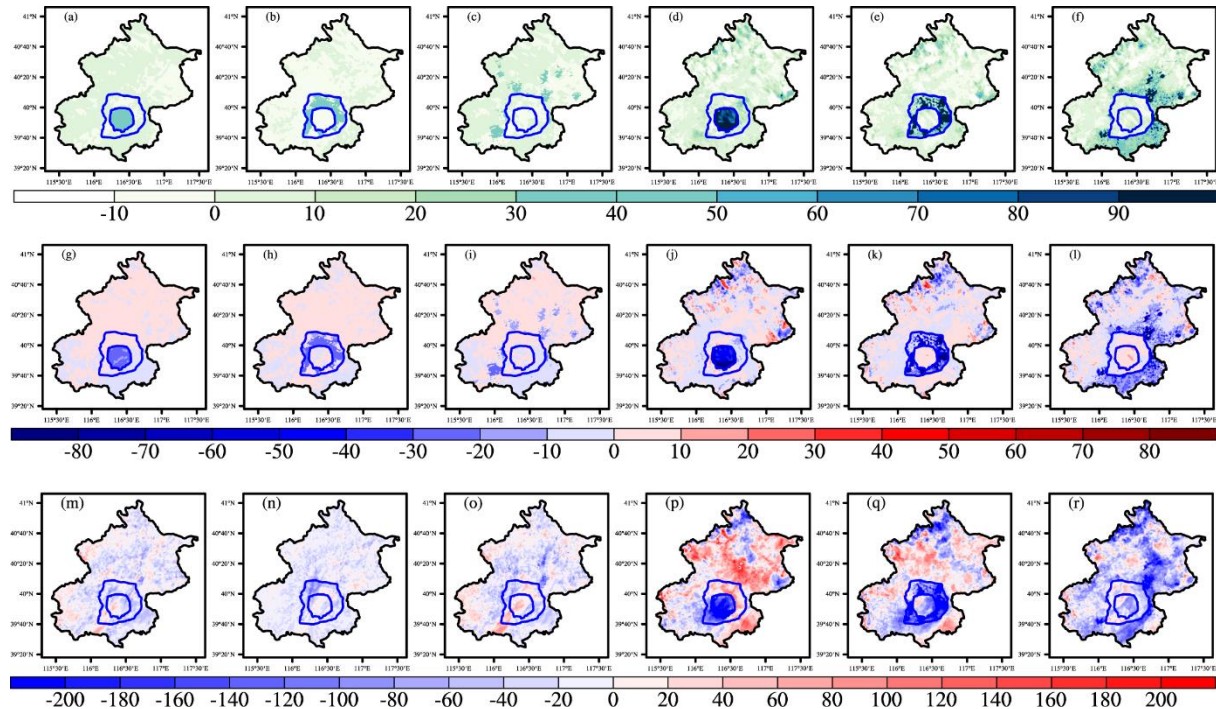

**Figure 11. Changes in the latent heat flux, sensible heat flux, and boundary layer height due to road sprinkling and urban irrigation. (a) Changes in latent heat flux (LH) due to road sprinkling in the city center. (b) Changes in LH due to road sprinkling in the suburbs. (c) Changes in LH due to road sprinkling in rural areas. (d) Changes in LH due to urban irrigation in the city center. (e) Changes in the LH due to urban irrigation in suburbs. (f) Changes in LH due to urban irrigation in rural areas. (g)–(l) Changes in sensible heat flux in a similar manner to (a)–(f). (m)–(r) Changes in the boundary layer height in a similar manner to (a)–(f).**

The quadratic functions fitted to the relationships between the amounts of water applied and the cooling effects in the simulations are shown in Figure 12 and Table 4. The normalized amounts of water applied in the city center, suburbs, and rural areas for road sprinkling and urban irrigation were $0.528 \times 10^8$, $0.2868 \times 10^8$, and $0.039 \times 10^8$, and $0.81 \times 10^8$, $1.72 \times 10^8$, and $4.45 \times 10^8 \, \text{m}^3/\text{month}$, respectively. The actual amounts of water applied can be determined by multiplying the values on the x-axis and those given above. The results showed that the cooling effects of road sprinkling were similar in the three parts of the city, where the temperature decreased by a maximum of about 0.55°C, and the cooling effects remained stable when more water was sprinkled on the roads in the suburbs and rural areas. Sprinkling the roads in the city center changed the regional temperature more rapidly compared with sprinkling water in the suburbs and rural areas. However, the cooling efficiency did not increase according

to the amount of water applied (for each normalized amount of water applied, the effect of water sprinkling in the city center was twice that that in the suburbs and 10 times than that in the rural areas,

but the cooling effect was similar, where the cooling efficiency was lower in the city center than suburb and rural areas), possibly because the water sprinkled in the city center was concentrated in a smaller area where the wind was reduced and this decreased the cooling efficiency. The cooling effect of urban irrigation was generally greater than that of road sprinkling and it was most significant in rural areas. However, the cooling efficiency of urban irrigation was lower in rural areas than the city center and

suburbs because in order to apply the same normalized amount of water for irrigation, the actual amount of water applied in rural areas was almost five times that in the city center and two times that in the suburbs, but the temperature decrease in the rural areas was only 1.5 times higher than those in the city center and suburbs. Thus, the effect of urban irrigation was most efficient in the city center. The cooling effects in the city center and suburbs increased as the amount of water applied increased, which differed

slightly from the effect of road sprinkling. We also found that some points deviated greatly compared with others in the city center under road sprinkling according to regression analysis (Figure 12(a)), possibly because road sprinkling was limited to a small area in the city center with a low amount, and the overall effect could not be determined based on the cooling of the city center due to the occurrence of random processes in the atmosphere. Also other researchers showed that cooling effect was various with

different water amount, regions and weather conditions (Broadbent et al., 2018; Wang et al., 2019; Gao et al., 2020).

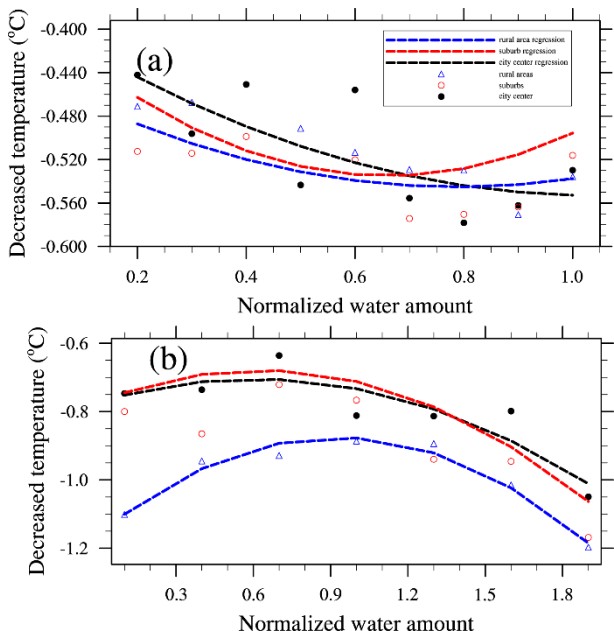

**Figure 12. Relationships between the amount of water applied and the cooling effect: (a) for road sprinkling and (b) for urban irrigation. The black dots denote the cooling effect of water applied in the city center based on model simulations, red circles denote the cooling effect of water applied in suburbs based on model simulations, and blue triangles denote the cooling effect of water applied in rural areas based on model simulations. The lines are polynomial regression curves, and black, red, and blue represent the city center, suburbs, and rural areas, respectively.**

**Table 4. Relationships between the amounts of water applied and cooling effects**

|  | Road sprinkling | Urban irrigation |
|---|---|---|
| City center | $f = 0.15 * w^2 - 0.32 * w - 0.39$ | $f = -0.18 * w^2 + 0.22 * w - 0.77$ |
| Suburb | $f = 0.34 * w^2 - 0.45 * w - 0.42$ | $f = -0.23 * w^2 + 0.30 * w - 0.86$ |
| Rural area | $f = 0.16 * w^2 - 0.26 * w - 0.44$ | $f = -0.33 * w^2 + 0.61 * w - 1.16$ |

**3.3 Optimal water use strategies**

The problem of distributing water to plants and roads in different parts of a city in order to obtain the optimal cooling effect can be solved using an optimization scheme. In this study, we divided the city into three parts according to the population density and urban type, i.e., the city center, suburbs, and rural areas. If the relationships between the amounts of water applied and the cooling effects were known for the three parts of city, then an optimal water usage scheme could be developed, where the optimization

objective could be defined as the comprehensive temperature decrease attributable to both road sprinkling and urban irrigation in the city center, suburbs, and rural areas. The optimal water usage scheme could be described as follows:

$$\text{Max: } \sum_{j=1}^{3}\sum_{i=1}^{2} f_{i,j}(w_{i,j}), \tag{5}$$

where i represents road sprinkling and urban irrigation, with i equal to 1 or 2, respectively; j represents the city center, suburbs, and rural areas, with j equal to 1, 2, and 3, respectively; and $f_{i,j}(w_{i,j})$ is a function of the normalized amount of water applied and the cooling effect, which can be fitted based on the model simulation results presented in Section 3.2.

Considering that the total amount of urban water supplied for road sprinkling and urban irrigation is a fixed value, the water demand for each part of city should satisfy the minimal needs for the municipal services and plants in terms of ecology, farmland, and roads. Thus, the constraint conditions for optimizing the usage of water are as follows:

$$\text{s.t.} \quad \sum_{j=1}^{3}\sum_{i=1}^{2} w_{i,j} = A, \tag{6}$$

$$b_j \ll w_{1,j} \ll B_j, j = 1,2,3, \tag{7}$$

$$c_j \ll w_{2,j} \ll C_j, j = 1,2,3, \tag{8}$$

where $A$ represents the total water supplied for road sprinkling and urban irrigation; $j$ represents the city center, suburbs, and rural areas, with $j$ equal to 1, 2, and 3, respectively; $b_j$ represents the minimal water demand for road sprinkling, $B_j$ represents the maximum water supply for road sprinkling, $c_j$ represents the minimal water demand for urban irrigation, and $C_j$ represents the maximum water supply for urban irrigation. In addition, other water restrictions applied in each part of the city can be expressed as other equalities or inequalities in this optimal water usage scheme.

Based on the historical water demand and supply for municipal services described in the Water Resources Bulletin of Beijing, the Requirements for Quality and Operation of City Road Sweeping and Cleaning, and the textbook entitled "Water Supply Engineering," we set the constraint conditions given in Table 5. In addition, according to Beijing's 13th Five Year Plan, the water supply for the whole city was set as $43 \times 10^8$ m$^3$/year and the total amount of water for road sprinkling and urban irrigation as about $17 \times 10^8$ m$^3$/year, which was mainly consumed in summer periods, excluding the water usage by

industry and residents. Solving this optimization problem started by artificially assigning initial values to the solution and then solving the initial objective function value in Eq. (5), before assigning new values to the solution according to the search algorithms and the constraint conditions in Eqs (6)–(8). A new objective function value was solved during this step. These two steps were repeated until the objective function value changed little. We used the Optimization Toolbox in MATLAB to solve the problem. After 24 loop iterations, the results showed that the normalized amounts of water applied for road sprinkling in the city center, suburbs, and rural areas were 0.4, 0.2, and 0.1 (the actual amounts of water applied were $0.21 \times 10^8$, $0.06 \times 10^8$, and $0.04 \times 10^7$ m$^3$/month, and road sprinkling in the whole city accounted for almost 10% of the total water supply), respectively, and the normalized amounts of water applied for urban irrigation in the city center, suburbs, and rural areas were 0.43, 0.36, and 0.40 (the actual amounts of water applied were $0.35 \times 10^8$, $0.62 \times 10^8$, and $1.78 \times 10^8$ m$^3$/month, respectively, and urban irrigation in the whole city accounted for almost 90% of the total water supply) to obtain the greatest cooling effect with a temperature decrease of 1.9°C. These results were reasonable because the enhanced cooling effect of road sprinkling decreased as the amount of water applied increased, so the lowest water demand was similar when more water was applied by road sprinkling. Urban irrigation in rural areas accounted for a large proportion of the total water supply in order to satisfy the needs for crop growth and environmental cooling.

**Table 5. Constraint conditions for water usage in urban, suburban, and rural areas**

| | RS-urban | RS-suburb | RS-rural | UI-urban | UI-suburb | UI-rural |
|---|---|---|---|---|---|---|
| Conversion factor ($10^8$ m$^3$/month) | 0.528 | 0.2868 | 0.039 | 0.81 | 1.72 | 4.45 |
| Lowest water demand | 0.4 | 0.2 | 0.1 | 0.1 | 0.1 | 0.1 |
| Highest water supply | 1 | 1 | 1 | 2 | 2 | 2 |

RS: road sprinkling, UI: urban irrigation

The optimized results may be slightly higher than the actual requirements because the water usage in one part of a city will affect the cooling effects in other parts. The uncertainties related to the constraint conditions will also affect the results. The total amount of water applied for road sprinkling and urban irrigation (A in Eq. (6)) must be considered among these uncertainties. Thus, we conducted sensitivity based on proportions ranging from 0.7 to 1.5 relative to $17\times 10^{8}$ m$^{3}$/year for the total amount of water applied. The results showed that the relationship between the total amount of water applied (A) and the cooling effect was nonlinear, where the temperature decreased sharply as the total amount of water applied (A) increased (see Figure 13).

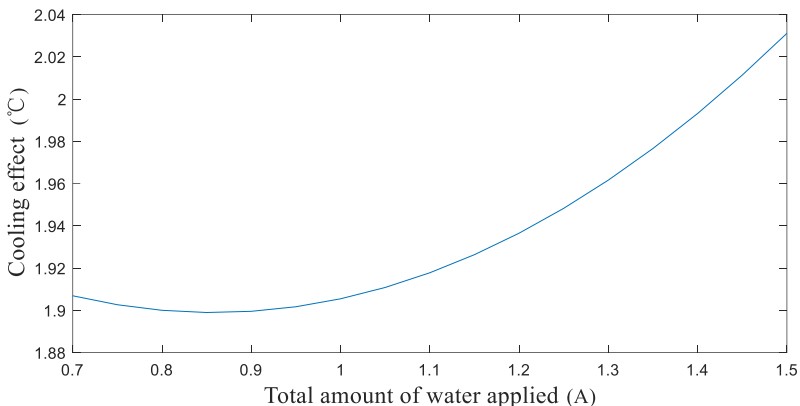

**Figure 13. Sensitivity analysis based on the total amount of water applied and the cooling effect.**

**4. Conclusions**

In this study, we coupled improved water usage schemes for road sprinkling and urban irrigation in the WRF model. The soil hydrological and urban canopy model were modified for this study. Simulations were then conducted at a resolution of 1 km where different amounts of water were applied via road sprinkling and urban irrigation in a case study set in Beijing, China. We determined the relationships between the amounts of water applied and the cooling effects in different parts of the city. The efficiency of the cooling effect due to road sprinkling decreased as the amount of water applied increased, where the city center cooled the most because more roads were present in a small area, whereas sprinkling had no significant effect in rural areas. Urban irrigation in the daytime cooled the city during the day and night because evapotranspiration from the plants and soil was enhanced. The cooling effect was more general

throughout the region and not limited to local areas. In addition, urban water usage locally increased the latent heat flux, but decreased the sensible heat flux and the boundary layer height. Some uncertainties were evident in the simulations. Different land use type data changed the urban and plant distributions, and conducting the simulations with the default MODIS-based land use data may have led to cooling effect errors. The direct driving factor responsible for the cooling effect was the amount of water used for these land use types and the differences were small. Obtaining better estimates of water use can reduce the errors due to land use data. In addition, the cooling effect in each part of the city was regionally averaged, which may have reduced the significance of specific land use types. However, these errors may increase in the future because of greater land use changes.

We also conducted an optimization process to determine the appropriate amounts of water for application by urban irrigation and road sprinkling in different parts of city, where we treated decreasing the temperature as the optimization objective and the total water supply, highest water supply in different parts of the city, and lowest water demands in the city center, suburbs, and rural areas as the constraint conditions. The optimization results showed that the temperature could be reduced by 1.9°C using road sprinkling and urban irrigation in the city center, suburbs, and rural areas when the normalized amounts of water are applied (i.e., 0.4, 0.2, and 0.1 for road sprinkling, and 0.43, 0.36, and 0.40 for urban irrigation, respectively). Sensitivity analysis based on the total water supply for the whole city (A) detected a nonlinear relationship between the total water supply (A) and the optimized decrease in the temperature, where the cooling effect increased sharply as the amount of water applied increased. Considering Beijing's 13th Five Year Plan, allocating about 90% of the total water to urban irrigation and 10% to road sprinkling is the most effective approach for mitigating high urban temperatures.

In addition, other large cities such as Tokyo, London, and Phoenix are affected by the threat of high temperatures in the summer. In these cities, urban water use management is an important part of municipal planning in order to balance the water demand and supply, as well as to improve the urban climate. Road sprinkling might not be a common solution for mitigating high temperatures in other countries but the optimal water usage scheme determined in the present study is still applicable to other cities, where the road sprinkling supply can be set to zero if no road sprinkling occurs.

*Code availability.* Model code can be obtained from the corresponding author, and the data used in this study are available from 4TU.Centre for Research Data (http://doi.org/10.4121/uuid:01621202-7ec4-4643-84b5-5f9ec2966004).

*Author contributions.* Bin Liu conducted the simulation analysis and prepared with manuscript.
Zhenghui Xie helped with manuscript preparation and editing. ChunXiang Shi provided the validation data. Shuang Liu and Yujing Zeng developed the initial model. Ruichao Li, Longhuan Wang, Yan Wang, and Si Chen helped with data analysis. Binghao Jia, Peihua Qin, and Jinbo Xie offered valuable suggestions.

*Competing interests.* The authors declare that they have no conflict of interest.

*Acknowledgments.* The authors thank the National Meteorological Information Center, China Meteorological Administration for data support.

*Financial support.* This study was supported by the Strategic Priority Research Program of Chinese Academy of Sciences (grant number XDA23090102), the National Natural Science Foundation of China (NSFC) project (grant number 41830967), the Key Research Program of Frontier Sciences, Chinese Academy of Sciences (CAS) (grant number QYZDY-SSW-DQC012), and the National Key Research and Development Program of China (grant numbers 2018YFC1506602 and 2020YFA0608203).

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
