# Peer review of "Optimal water use strategies for mitigating high urban temperatures"

_Hydrology and Earth System Sciences, 2020_

## Referee Comment (RC1) · Anonymous Referee #1 · 2 Jul 2020

Recent urbanization has been changing the regional climate significantly, and the urban irrigation and road sprinkling not only make the land surface processes more complex, but also influence the urban canopy temperature. Bin Liu et al. did a novel job to optimize the limited water supply between urban irrigation and road sprinkling to get the maximum cooling effects over the Beijing city. The authors introduced the urban irrigation and road sprinkling in the WRF model, estimated the cooling efficiency of urban irrigation and road sprinkling over the urban, suburban and rural areas, and developed an optimal water management scheme to get the largest cooling effects with limited water resource. However, there are still some issues should to be revised before its publication on the HESS journal.

Major comments: 1.Is it appropriate to include the rural areas (outside the sixth ring

road) in the analysis? Although the authors treat the "urban irrigation" as ecological and farmland irrigation, the farmland irrigation in the rural region seems only cools the rural temperature with little influence on the urban temperature. 2.The authors did a good job in introducing the optimal water usage scheme and model development, but the description of the experimental design is confusing. For example, the authors said "Three experiments were conducted to consider no water usage, urban irrigation and road sprinkling". Does it mean only one experiment was conduced with consideration of road sprinkling? If so, how can they separate the cooling effects of urban road sprinkling on the suburban and rural areas (Fig. 8a)? In addition, what does the "A climate summer time periods from 2000 to 2017 were averaged to 4 days which represent the climatic May, June, July and August. And the first day was considered as the spin up period. " mean? Does this mean for each month, only one day simulation forced by climatic boundary condition is performed? 3.The offline experiment using CLM4.5 model was used to "illustrate the cooling effect of urban irrigation and road sprinkling". Does the author also choose the CLM4.5 model in the WRF modeling? Moreover, the offline modeling shows that urban irrigation does not influence the latent/sensible heat significantly over the urban region (within the fifth ring road; Figs. 5a,5c), but the online modeling shows contrary result where clear influence of urban irrigation over urban region (Figs. 10a,10d). How to interpret this? 4.Can the default USGS land use category in the WRF model represent the urban land use in the research region? As is shown in Figure 3, the land use type in the Beijing city is 12~14. But the urban land use type in USGS category is 1. 5.The current optimization method does not consider the urban extension or land cover change in the future. The author should at least discuss the influence of this neglect on the result.

Minor comments: 1.L134, change "in-situ" to "In-situ" 2.I suggest the authors to give a short description of how to estimate the road sprinkling in section 2.3 and a plot of road sprinkling water use in Beijing in Figure 2. 3.In Table2, the land surface model option is "CLM/NOAH-MP", does this mean the authors performed ensemble simulation using different land surface models? 4."The simulation results showed that urban irrigation

decreased the water table depth due to groundwater extraction ". Why does the water table depth decrease? If the water is extracted, the water table depth will increase (e.g., from 4m to 5m), and the difference is positive. 5.L205-L215. The author evaluate the WRF simulation by using CLDAS and observation. Does the WRF simulation used here consider the urban irrigation and road sprinkling? And will the incorporation of the above two processes have some improvements on the temperature simulations?

---

## Referee Comment (RC2) · Anonymous Referee #2 · 2 Jul 2020

This paper aims to determine an optimal water use strategy for urban cooling in Beijing by coupling a novel water use scheme into WRF and conducting summertime simulations. The topic does read interesting and the paper well fits the scope of HESS; however, the presentation of current manuscript, which has great room to improve, does hinder the my understanding of some key points of this work.

As such, the authors are suggested to improve the manuscript by considering the following concerns:

1. As numerous WRF simulations have been done in Beijing, I have less concerns about the model performance per se; instead, I would encourage the authors to investigate more if the consideration of urban water use could effectively improve the WRF simulations in Beijing.

[Figure]

2. It is very unclear how the optimisation of water use is done in section 3.3, which, however, should be one of the key contributions the paper attempted to make. I suggest to incorporate sections 2.1 into 3.3, so the optimal strategy part could be more coherently presented.

3. Presentation is a major issue: grammatical errors and typos are pervasive; figures in general miss appropriate caption and proper legends. A small portion of the issues are given below as examples: - L58: "role" –> "roles" - L101: "nonmatter" –> "no matter" - L115: "he specific" –> "the specific" - caption of Figure 1: "it's coupling" –> "its coupling"; the full name of WRF is unnecessarily given twice; - caption of Figure 4: "moister" –> "moisture". - units: watts should be in uppercase, i.e., "W". - in English, the words and punctuations should be separated by spaces. e.g., in "model(CLM4.5)", a space should be added between "model" and "(". - ...

Other specific comments:

1. Figure 1: please explain: - What is 'time judge'? - Why is the road sparkling only activated during summer night? And when is summer? When is night? (when kdown==0?) - What is "imperative layer" in the "Urban Canopy" circle of panel b?

2. Section 2.3: Unclear what experiments were actually carried out in this work.

3. L165: Please provide more details on the construction of 4-day climate ensemble.

4. L175: This term "atmospheric stochastic processes" is confusing: either provide examples or reword it.

5. Figure 11a: the regression for results of city centre looks problematic.

6. Table 4: it's unclear what the "units" row (the first row) indicate.

---

## Author Comment (AC1) · 30 Jul 2020

We would appreciate all the constructive comments by the anonymous referees. We have substantially revised the paper and improved the English expression. All the modification can be found in the revised manuscript. The responses in detail to RC1 are listed below.

Recent urbanization has been changing the regional climate significantly, and the urban irrigation and road sprinkling not only make the land surface processes more complex, but also influence the urban canopy temperature. Bin Liu et al. did a novel job to optimize the limited water supply between urban irrigation and road sprinkling to get the maximum cooling effects over the Beijing city. The authors introduced the urban irrigation and road sprinkling in the WRF model, estimated the cooling efficiency of urban irrigation and road sprinkling over the urban, suburban and rural areas, and developed an optimal water management scheme to get the largest cooling effects with limited water resource. However, there are still some issues should to be revised before its publication on the HESS journal. Major comments:

**Q1:** Is it appropriate to include the rural areas (outside the sixth ring road) in the analysis? Although the authors treat the "urban irrigation" as ecological and farmland irrigation, the farmland irrigation in the rural region seems only cools the rural temperature with little influence on the urban temperature.)

**A:** we think it is better to take rural areas into consideration because (1) total amount of water applied (including rural, suburb and city center) is one of known constraint conditions, and it is not easy to separate those water; (2) We take the whole city as an optimal objective, rural areas cannot be considered separately when applying optimization method; (3) the rural urban irrigation has influence to other parts of city, although smaller than non-rural area.

**Q2:**

■ (1) the description of the experimental design is confusing. For example, the authors said "Three experiments were conducted to consider no water usage, urban irrigation and road sprinkling". Does it mean only one experiment was conducted with consideration of road sprinkling? If so, how can they separate the cooling effects of urban road sprinkling on the suburban and rural areas (Fig. 8a)?

**A:** They are three kinds of experiments actually, not only 3 experiments (total 1+21+27=49 experiments). they are (1) no water usage experiment includes no urban irrigation and no road sprinkling in city center, suburb and rural areas; (2) urban irrigation experiments include 21(7x3) individual experiments, they are 0.1 to 1.9 times (0.1, 0.4, 0.7, 1, 1.3, 1.6, 1.9) of the estimated urban irrigation among city center, suburb and rural areas separately. (3) Road sprinkling experiments include 27(9x3) individual experiments, they are road sprinkling in city center, suburb and rural areas separately, water amount ranged from 0.2 to 1 times (0.2, 0.3, 0.4, 0.5,0.6,0.7,0.8,0.9,1) of the maximum water-holding capacity. A new table added to detail this, see Table 2, L144.

**Table 1. Descriptions of experiments designs**

| Experiments | Area | Water amount | Descriptions |
|---|---|---|---|
| Raw experiment | / | / | No urban irrigation and no road sprinkling |
| Urban irrigation experiments | City centers | 0.1, 0.4, 0.7, 1, 1.3, 1.6 and 1.9 times of the estimated urban irrigation in each part of city | Urban irrigation in city center with different water amount |
| | Suburb areas | | Urban irrigation in suburb areas with different water amount |
| | Rural areas | | Urban irrigation in rural areas with different water amount |
| Road sprinkling experiments | City centers | 0.2, 0.3, 0.4, 0.5, 0.6, 0.7, 0.8, 0.9 and 1 times of the maximum water-holding capacity of impervious layer | Road sprinkling in city center with different water amount |
| | Suburb areas | | Road sprinkling in suburb areas with different water amount |
| | Rural areas | | Road sprinkling in rural areas with different water amount |

Fig. 8a shows the cooling effect of city center in the condition of 0.5 times of the maximum water-holding capacity.

■ (2) In addition, what does the "A climate summer time periods from 2000 to 2017 were averaged to 4 days which represent the climatic May, June, July and August. And the first day was considered as the spin up period. " mean? Does this mean for each month, only one day simulation forced by climatic boundary condition is performed?

A: Yes. Firstly, we found out all data for May from 2000 to 2017. Then, averaged all these data to one day which represented climatic May. At last, climatic June, July and August were got by repeating above two steps. We did this to save simulation time. Revised in L152-153.

**Q3:**

■ The offline experiment using CLM4.5 model was used to "illustrate the cooling effect of urban irrigation and road sprinkling". Does the author also choose the CLM4.5 model in the WRF modeling?

A: The offline simulation is to validate the urban water usage scheme (urban irrigation and road sprinkling) , (here we just take CLM4.5 as an example). It proves that taking urban water usage scheme into land surface model is better (no matter which land surface model). In WRF model, we didn't use CLM4.5 because 1) WRF does not couple the complete CLM model 2) the plant function types are different between offline (CLM in CESM) and coupling model (WRF) , so we choose SLUCM in coupling simulation with the same urban water usage schemes (the same in CLM).

■ (2) Moreover, the offline modeling shows that urban irrigation does not influence the latent/sensible heat significantly over the urban region (within the fifth ring road; Figs. 5a,5c), but the online modeling shows contrary result where clear influence of urban irrigation over urban region (Figs. 10a,10d). How to interpret this?

A: Urban irrigation has the influence, with little influence (even no) influence within the 3[rd] ring road, but the influence between 4[th] to 5[th] ring road (including some part of 3[rd] ring road) are strong.

The difference between offline and coupling model are (1) subgrid type of offline model within 3$^{rd}$ ring road are mostly urban and little pfts (plant function types), and impervious layers, walls and roofs are major parts in urban without evaporation. Besides, the estimated water amount within 3$^{rd}$ ring road are less, so influence of urban irrigation was not so significant (especially with 3$^{rd}$ ring road); (2) offline simulations had no interactive effect between atmosphere and land surface, but online simulation had which influence some variables over larger region.

**Q4:** Can the default USGS land use category in the WRF model represent the urban land use in the research region? As is shown in Figure 3, the land use type in the Beijing city is 12~14. But the urban land use type in USGS category is 1.

A: I recheck the data, they were MODIS-based Land Use Classifications, I mix them up in this manuscript. The cooling effect mostly driven by water use amount. The simulation with default land use type (not the land use type of 2000-2017) may bring some uncertainty; I have discussed the uncertainty in the last section. See L333-338.

**Q5:** The current optimization method does not consider the urban extension or land cover change in the future. The author should at least discuss the influence of this neglect on the result.

A: I discussed the uncertainty in the last section. See L333-338.

Minor comments:
**Q1:** L134, change "in-situ" to "In-situ"
A: Revised

**Q2.** I suggest the authors to give a short description of how to estimate the road sprinkling in section 2.3 and a plot of road sprinkling water use in Beijing in Figure 2.
A: Water amount of road sprinkling is proportional to the maximum water-holding capacity (Eq. 1) in road sprinkling scheme. So, spatial distribution of road area proportion was added in Figure 2(b).

[Figure]

**Figure 1. (a) Estimated urban irrigation water use in Beijing (mm/s), (b) spatial distribution of road area proportion (%).**

**Q3.**In Table2, the land surface model option is "CLM/NOAH-MP", does this mean the authors performed ensemble simulation using different land surface models?

A: Revised. See Table 3

**Q4.**"The simulation results showed that urban irrigation decreased the water table depth due to groundwater extraction " . Why does the water table depth decrease? If the water is extracted, the water table depth will increase (e.g., from 4m to 5m), and the difference is positive.

A: It's water table of ground not water table depth. The two variables are opposite. Revised in L165.

**Q5.**L205-L215. The author evaluate the WRF simulation by using CLDAS and observation. Does the WRF simulation used here consider the urban irrigation and road sprinkling? And will the incorporation of the above two processes have some improvements on the temperature simulations?

A: We validate the schemes in offline model (we take CLM as an example), it improves in sensible/latent heat flux (see Figure 6). In WRF simulation we drew the spatial distribution of correlation coefficients between simulation and CLDAS data in Figure 7.

---

## Author Comment (AC2) · 30 Jul 2020

We would appreciate all the constructive comments by the anonymous referees. We have substantially revised the paper and improved the English expression. All the modification can be found in the revised manuscript. The responses in detail to RC2 are listed below.

This paper aims to determine an optimal water use strategy for urban cooling in Beijing by coupling a novel water use scheme into WRF and conducting summertime simulations. The topic does read interesting and the paper well fits the scope of HESS; however, the presentation of current manuscript, which has great room to improve, does hinder the my understanding of some key points of this work.

As such, the authors are suggested to improve the manuscript by considering the following concerns:

**Q1:** As numerous WRF simulations have been done in Beijing, I have less concerns about the model performance per se; instead, I would encourage the authors to investigate more if the consideration of urban water use could effectively improve the WRF simulations in Beijing

**A:** we have validated urban water usage scheme in offline model (take CLM as an example), it shows better results (latent/sensible heat flux) in Figure 6, this process cannot be missed in land surface model.

**Q2**. It is very unclear how the optimisation of water use is done in section 3.3, which, however, should be one of the key contributions the paper attempted to make. I suggest to incorporate sections 2.1 into 3.3, so the optimal strategy part could be more coherently presented.

**A:** We revised this manuscript as suggested, and detail the part of optimization. See section 3.3

**Q3**. Presentation is a major issue: grammatical errors and typos are pervasive; figures in general miss appropriate caption and proper legends. A small portion of the issues are given below as examples: - L58: "role" –> "roles" - L101: "nonmatter" –> "no matter" - L115: "he specific" –> "the specific" - caption of Figure 1: "it's coupling" –> "its coupling"; the full name of WRF is unnecessarily given twice; - caption of Figure 4: "moister" –> "moisture". - units: watts should be in uppercase, i.e., "W". - in English, the words and punctuations should be separated by spaces. e.g., in "model(CLM4.5)", a space should be added between "model" and "(". - ...

**A**: Errors above have been revised.

**Q4:**

**(1)** Other specific comments: Figure 1, What is 'time judge'? - Why is the road sparkling only activated during summer night? And when is summer? When is night? (when kdown==0?) - What is "imperative layer" in the "Urban Canopy" circle of panel b?

**A:** According to "Cleaning quality and operation requirements of urban road cleaning (DB11/T 353-2014)" (revised in L114), road sprinkling should be finished before 5:00 a.m. I think this action is to avoid disturbing traffic. although road sprinkling was seen in daytime.

In the model we can judge summer and night according to model time.

imperative layer here mainly means the imperative road in this manuscript. Description added in L105.

**(2)** Section 2.3, Unclear what experiments were actually carried out in this work.

A: A new table added to detail the experiments and more description in section 2.2. See Table 2

from L144.

**Table 1. Descriptions of experiments designs**

| Experiments | Area | Water amount | Descriptions |
|---|---|---|---|
| Raw experiment | / | / | No urban irrigation and no road sprinkling |
| Urban irrigation experiments | City centers | 0.1, 0.4, 0.7, 1, 1.3, 1.6 and 1.9 times of the estimated urban irrigation in each part of city | Urban irrigation in city center with different water amount |
| | Suburb areas | | Urban irrigation in suburb areas with different water amount |
| | Rural areas | | Urban irrigation in rural areas with different water amount |
| Road sprinkling experiments | City centers | 0.2, 0.3, 0.4, 0.5, 0.6, 0.7, 0.8, 0.9 and 1 times of the maximum water-holding capacity of impervious layer | Road sprinkling in city center with different water amount |
| | Suburb areas | | Road sprinkling in suburb areas with different water amount |
| | Rural areas | | Road sprinkling in rural areas with different water amount |

**(3)** L165: Please provide more details on the construction of 4-day climate ensemble

A: Revised, see L152-154. "A climate summer time periods from 2000 to 2017 were averaged to 4 days which represent the climatic May, June, July and August. Firstly, we found out all data for May from 2000 to 2017. Then, averaged all these data to one day which represented climatic May. At last, climatic June, July and August were got by repeating above two steps."

**(4)** L175, This term "atmospheric stochastic processes" is confusing: either provide examples or reword it.

A: reword to "random processes in the atmosphere" L160

**(5)** Figure 11a: the regression for results of city center looks problematic

A: I rechecked them, it's correct, it showed like that because one-point deviates from other points larger.

**(6)** Table 4: it's unclear what the "units" row (the first row) indicate

**A:** the first raw are conversion factors. We regard the estimated water amount in each part of city (city center, suburb and rural area) as the standard one unit. The optimized results are standard values, so the actual water amount can be got by multiply the conversion factors. To avoid misunderstanding we reword to "conversion factor". See Table 4 in L336

---

## Author Response (AR2)

**We appreciate all the constructive comments by anonymous referee and editors. We have substantially revised the paper and improved the English expression. The responses in detail are listed below.**

**Q1:** About no answering **"**As numerous WRF simulations have been done in Beijing, I have less concerns about the model performance per se; instead, I would encourage the authors to investigate more if the consideration of urban water use could effectively improve the WRF simulations in Beijing"
**A:** More WRF simulation with and without water usage schemes has been conducted. The results can be seen in Sec. 3.1 (from L244 and Figure8)

**Q2:** About "Grammatical errors and typos, L166 etc."
**A:** Revised

**Q3:** About "(DB11/T353-2014)"
**A:** It's the regulation number of Requirements for Quality and Operation of City Road Sweeping and Cleaning. A web link was added as a reference. (L105)

**Q4** About the question "However, I doubt if such an ensemble is appropriate for this simulation: 1) how did you deal with rainfall? It's highly possible each hour of the 18 years' summer may experience whatever amount of rainfall; so in your ensemble day, it's raining through a day? 2) how did you deal with wind regimes? If southerly and northerly winds with the same speed happened in the same hour, would it end up with a zero wind speed in your ensemble day?"
**A:** We aim to find out the relationships between water amount and cooling effect from ensemble lateral boundary conditions, rather than to simulate the real cases. We think it is acceptable to do those simulations.
1) on some level, the rain can be regarded as urban water use (or we can say urban water usage can be regarded as rain). Because (1) they have the same physical process once they dropped to the ground in model. (2) road sprinkling always happen, no matter rain or not (according to Quality and Operation of City Road Sweeping and Cleaning). (3) the ensemble rainfall is small

2) Horizontal wind represented by vector (U,V). In your case, it's zero. However, in the ensemble day, (U,V) were averaged monthly and yearly. and the wind direction of Beijing in summer is normally Southeast, and it can hardly be zero in your case.

**Q5:** About "Even though the 2nd order polynomial regression may appear as so, my actual concern is the applicability of this regression function, which was later used in your optimisation:Apparently, rather than "one point deviates from other points" as you claimed, points with x=0.4, 0.6 and 1.0 deviate from the regression. And the deviation is in fact intriguing: why would the increase in water amount reduce the cooling effect at certain points?Without addressing this concern, the optimisation-related analysis might not be justified."
**A:** We guess two reasons may case this problem, 1) the randomly process in model's running, 2) road sprinkling to city center was limited in a small area with less water

amount, and the whole effect cannot be determined by cooling of city center (road sprinkling in city center has strong impact on city center, but small uncertainty in rural area has more influence to whole city, that may be the reason why regression of city center was so different once the random processes in the atmosphere happened).

It may better if we do ensemble simulations with different initial conditions and different physical schemes. Or we can exclude the abnormal values. But we didn't do that because 1) actually, the experiments are relatively ideal experiments (mentioned in Q4). 2) For optimal analysis, the past/future relationships between water amount and cooling effect remain uncertain. 3) the main purpose is to construct a method of optimal water use strategies; we show what we get. And we add a discussion about the uncertainties in Sec.3.2 (L324)

---

## Author Response (AR3)

We appreciate all the constructive comments by reviewer and editors. We have revised the paper according to the comments of reviewer, also we revised some other errors. The responses in detail are listed below.

样式定义: 正文

- A1: Thanks for the added simulation results. For the panels c and d in new Figure 8 from this revision, what is the difference from Figure 6? Based on the caption, I consider they should be the same; however, the figures are showing different results. Please be specific and revise the captions of both figures.

RE:Thanks for your comment, the captions of the two figures were revised and they were specific now. (L225: Figure 6. (a) Comparisons of sensible heat flux and (b) latent heat flux according to station observations and CLM simulations. Blue dots are the raw CLM simulation results and orange dots are the CLM simulation results with urban water usage scheme. L260: Figure 8. (c) sensible heat flux comparisons between station observation and WRF simulation, (d) latent heat flux comparisons between station observation and WRF simulation. Blue dots are the raw simulation results and orange dots are the simulation results with urban water usage schemes.).

- A2: it is a pity that now even in A5 of the response several grammatical errors/typos appear:
- "two reasons may case ..." --> "two reasons may cause"
- "It may better ..." --> "it may be better"
I won't comment more on this issue but just feel the response was prepared less seriously.

RE:Thanks for your comments. Some errors of the manuscript have revised, too.

- A3: thanks for the addition.
RE: Thanks.

- A4: regarding (2), I can accept the answer, which is less satisfactory than expected though. It would be better to show a distribution plot of wind speed so my previous question can be more convincingly addressed in a quantitative way

RE: Thanks for your suggestion. We added the figure of surface wind speed (the first layer of the wind which near the surface mostly) here. According to the figure, the wind speed is not zero. It means the question you mentioned (If southerly and northerly winds with the same speed happened in the same hour, would it end up with a zero wind speed in your ensemble day) would not happen.

[Figure]

Figure 1. (a)-(c) surface wind speed of climatic June, July and August.

- A5:research is in a sense about guessing but should not stop there: you put up the hypotheses, and you also need
to either accept or reject them. I understand these might be challenging but at least some attempts should be
35 conducted and presented, in particular for hypothesis 2. Although a discussion has been added in sect. 3.2, it is
more like admission of the limitation rather than quantitative analysis of its impact: if the regression relationships
here are invalid, the following analysis and implications in sect 3.3 are rather shaky and unconvincing.

RE: Thanks for your comments. we admit the limitation of this work, and there are many uncertainties in the
manuscript including model simulations and analysis. Considering that the goal of this manuscript is to construct a
40 optimal water usage method, we didn't do deeper analysis about the question of the scale water applied or randomly
process in model simulation.

Some references were about the differences of cooling effect, which may be related to the question (L330)
[Other researchers showed that cooling effect was various with different water amount, regions and weather
conditions (Broadbent et al., 2018; Wang et al., 2019; Gao et al., 2020)] so far. Besides, we are considering to write
45 another paper to illustrate the question with more specific simulations and analysis.

Broadbent A M, Coutts A M, Tapper N J, Demuzered M: The cooling effect of irrigation on urban microclimate during
heatwave conditions, Urban climate, 23, 309-329, 2018.
Gao K, Santamouris M, Feng J: On the cooling potential of irrigation to mitigate urban heat island. Science of The Total
50 Environment, 139754, 2020.
Wang C, Wang Z H, Yang J: Urban water capacity: Irrigation for heat mitigation. Computers, Environment and Urban
Systems, 78, 101397, 2019.

Major change list:
1. Captions of Figure6 and Figure8 (L225, L260).
55 2. Discussions in Sec.3.2 and adding references (L330).
3. Some errors and English expression.

[revised manuscript text omitted]

---

## Author Response (AR4)

**Changes**

1. Figure adjustment including figure size, font size, unit and so on.
2. Minor revision of the caption (Figure 8, L264)

[revised manuscript text omitted]